# Pysanky to Microfluidics: An Innovative Wax-Based Approach to Low Cost, Rapid Prototyping of Microfluidic Devices

**DOI:** 10.3390/mi15020240

**Published:** 2024-02-05

**Authors:** Philip J. Schneider, Liam B. Christie, Nicholas M. Eadie, Tyler J. Siskar, Viktor Sukhotskiy, Domin Koh, Anyang Wang, Kwang W. Oh

**Affiliations:** 1SMALL (Sensors and Micro Actuators Learning Lab), Department of Electrical Engineering, University at Buffalo, The State University of New York (SUNY at Buffalo), Buffalo, NY 14260, USA; philschn@buffalo.edu (P.J.S.); liamchri@buffalo.edu (L.B.C.); viktorsu@buffalo.edu (V.S.); dominkoh@buffalo.edu (D.K.); anyangwa@buffalo.edu (A.W.); 2Department of Mechanical Engineering, University at Buffalo, The State University of New York (SUNY at Buffalo), Buffalo, NY 14260, USA; nmeadie@buffalo.edu; 3Department of Physics, University at Buffalo, The State University of New York (SUNY at Buffalo), Buffalo, NY 14260, USA; tsiskar@buffalo.edu; 4Department of Biomedical Engineering, University at Buffalo, The State University of New York (SUNY at Buffalo), Buffalo, NY 14260, USA

**Keywords:** wax-based printing, prototyping, micromachining

## Abstract

A wax-based contact printing method to create microfluidic devices is demonstrated. This printing technology demonstrates a new pathway to rapid, cost-effective device prototyping, eliminating the use of expensive micromachining equipment and chemicals. Derived from the traditional Ukrainian Easter egg painting technique called “pysanky” a series of microfluidic devices were created. Pysanky is the use of a heated wax stylus, known as a “kistka”, to create micro-sized, intricate designs on the surface of an egg. The proposed technique involves the modification of an x-y-z actuation translation system with a wax extruder tip in junction with Polydimethysiloxane (PDMS) device fabrication techniques. Initial system optimization was performed considering design parameters such as extruder tip size, contact angle, write speed, substrate temperature, and wax temperature. Channels created ranged from 160 to 900 μm wide and 10 to 150 μm high based upon system operating parameters set by the user. To prove the capabilities of this technology, a series of microfluidic mixers were created via the wax technique as well as through traditional photolithography: a spiral mixer, a rainbow mixer, and a linear serial dilutor. A thermo-fluidic computational fluid dynamic (CFD) model was generated as a means of enabling rational tuning, critical to the optimization of systems in both normal and extreme conditions. A comparison between the computational and experimental models yielded a wax height of 57.98 μm and 57.30 μm, respectively, and cross-sectional areas of 11,568 μm^2^ and 12,951 μm^2^, respectively, resulting in an error of 1.18% between the heights and 10.76% between the cross-sectional areas. The device’s performance was then compared using both qualitative and quantitative measures, considering factors such as device performance, channel uniformity, repeatability, and resolution.

## 1. Introduction

Pysanky, derived from the verb pysaty, meaning to write, is the Ukrainian handcrafting tradition of creating intricate decorations and designs on Easter eggs using a wax resist method [1]. The process involves using a variety of wax styluses, also known as Kistkas, to draw designs and patterns on an egg. From there, the eggs are dipped in different dyes and then repainted over with wax, creating a multi-colored pattern. Once completed, the wax is removed, leaving behind a miniature, intricate, multi-colored, patterned egg, an example of which is shown in Figure 1a. A more detailed description can be read in the Appendix A (Appendix A
Appendix A). This inspiration created the foundation for this work. Using the same wax-egg design technique, a wax-based microfluidic device was fabricated and tested.

There are currently a number of different fabrication methods that are widely used to create microfluidic devices, including photolithography [2], laser etching [3], plasma treatment [4], micromachining [5], 3D printing [6,7,8,9], and so on [10]. However, the common limitations of these methods are that they often have time-consuming fabrication processes and require expensive instruments such as lithographic machines, lasers, 3D printers, plasma oxidizers, vacuum ovens, and sometimes clean rooms. Moreover, trained personnel are often required to operate such equipment. The aim of this study is to develop a low-cost, rapid prototyping, simple technique that can be used as a substitute for the complex microfabrication processes. This technique could potentially be used in both a research and educational setting, as well as in developing areas without the means and equipment to design, create, and test microfluidic chips.

Wax patterning, or wax-based microfluidics, consists of either constructing hydrophobic barriers using the wax or creating a mold out of the wax design [11,12]. Wax is often a chosen medium as it is inexpensive, simple [13], non-toxic, and dissolvable, opening up a range of applications from microfluidic mixers [14], microvalves [15,16], 3D micro-models [17], and so on. As we will discuss micromixers more in depth, as that was the preferred application for device fabrication validation, it is worth elaborating on the inherent advantages wax provides in the space of microvalves. The novel use of wax in microvalves has been demonstrated in other studies. Due to wax’s inherent physical properties, wax can be used as a leak-proof seal in a microfluidic application. Given its ability to undergo a phase change when introduced to heat or cold, wax can be a reversible microvalve that can handle pressures up to 50 psi.

Typically, wax-based microfluidics is complementary to that of a paper substrate, creating what are known as microfluidic paper-based analytical devices (μPADs) [18,19,20]. While these devices are typically substandard in terms of resolution and channel smoothness compared to those created using techniques like photolithography, they are significantly cheaper and use non-toxic reagents [21]. Some examples of wax patterning techniques include: wax dipping [22], wax screen printing [23,24,25,26], and wax printing and casting [27,28,29,30]. Wax dipping is a fabrication technique that uses a laser cutter to create an iron mold, which is then dipped into molten wax. Wax screen-printing uses solid wax to push through a patterned screen into the paper substrate, which is then placed on a hot plate to melt the wax. After which, the wax diffuses through the paper, forming hydrophobic barriers. Finally, there is wax printing, which most closely resembles the proposed technique here. Wax printing is where a sheet of paper is fed into a wax printer and is then processed by a heat cycle. While both wax dipping and screen printing are low-cost approaches, they have the disadvantage of being nonflexible in patterning and have low repeatability between device designs [21]. Very similar to that of an ink jet printer, these previous techniques can yield tens to hundreds of devices within a few minutes. However, the negative of this technique is how the wax is extruded from the print nozzle, a droplet-based printing method that results in irregularities in the channel dimensions [14], as well as the substrate limitation being only paper. Both factors limit the resolution and sharpness of the device.

This paper introduces a novel technique of wax printing using a low-cost, rapid prototyping and fabrication method similar to that of fused deposition modeling (FDM) 3D printing [7], in which an extruder tip translates in the x, y, and z axes to build up a model. The system combines this wax-based contact printing system with a heating and cooling substrate bed, controlled printing parameters, and the PDMS molding technique to create microfluidic devices (Figure 1). This wax acts as a sacrificial masking layer, which creates the imprint mold in the PDMS. By having a flexible, robust system, this technique overcomes many of the traditional limitations associated with wax-based printing. System advantages include open-source software, the ability to use different printing substrates, rapid prototyping capacities, and controllable operating parameters that yield feature sizes exceeding those of similar wax printing techniques. It should be noted that there are inherent advantages to this wax-based technique compared to traditional FDM printing. With this approach, we leverage the inherent natural properties of beeswax, from its viscosity at various temperatures to its solvability in various solutions to its ability to adhere to both 2D and 3D glass substrates.

Microscale fluidic flow models have proven valuable in the analysis of physical microfluidic devices [31]. To complement the real-world wax system, a 3D thermo-fluidic CFD model was developed. This model solves a series of Navier–Stokes and heat transfer equations for low-viscosity wax deposition, considering the fluid flow and solidification of the material as it spreads and cools on a substrate. This model allows for more extreme and unique prototyping to occur at a much faster and more inexpensive rate. For example, physical parameters such as substrate temperature, extruder tip temperature, viscosity of wax, surface finish, solidification dynamics, and their effect on the printed wax ridge can be easily viewed on the model. This initial model can be used to predict the resolution, uniformity, and functionality of the wax printing system, which in turn will allow for the exploration of new applications of the system. The initial results discussed are a preliminary simulation representation of the current wax printing setup in its most basic form.

## 2. Technology Overview

The underlying novelty of the enabling technology behind these wax-based microfluidic devices is a modified calligraphy machine (AxiDraw, Evil Mad Scientist, Sunnyvale, CA, USA; see Appendix A and video clip links for more details). In its original state, the user would attach a writing utensil device at the end of it, program in the design, and the AxiDraw would then proceed to write out that design on a surface. This machine was then retrofitted with a kistka tip and an electronic heating element, allowing for variable temperatures of the wax to be had, thus altering the viscosity and, as such, the flow of the wax onto the substrate. It should be noted that the wax in use here is 100% pure beeswax. Both the homogeneity and the mechanical properties of the wax are important for this study for both simulation purposes and to understand how these wax properties impact things like viscosity and wax flow rate, among other parameters. A detailed table of the wax properties used for both the simulation and calculations can be seen in Table 1. A 3D-printed slide holder was created for the substrates.

For prototyping purposes, 50.8 mm by 76.2 mm borosilicate glass slides were used. This system is shown in Figure 2.

A fluidic heating and cooling system was created to control the temperature of the print substrate, in this case, a glass slide. The temperature ranges from 2 °C to 51.7 °C. The temperature variation was performed by using a portable fluidic heating and cooling unit (ThermaZone Continuous Thermal Therapy, Innovate Medical Equipment, Cleveland, OH, USA). Figure 3 shows the thermal images of the system using a thermal infrared camera (FLIR Pro, FLIR Thermal Cameras, Wilsonville, OR, USA). Figure 3a shows the printing bed being heated to 37.5 °C. Even heating and cooling were not obtained as shown by the color gradients in the picture due to a number of prototyping flaws in the system, such as uneven flow distribution from the pump and varying thicknesses of the hollowed acrylic board at different points. Figure 3b shows the substrate bed being cooled. Here, a glass slide was placed on the printing bed and measured. As expected, the slide was not at the same temperature on the board, and as such, while performing experiments, the temperature of the glass slide was measured each time. Figure 3c shows the inlet ports connecting the cooling/heating pumps. Figure 3d shows the thermal image of the electronic heated kistka extruder tip.

In terms of software, the system runs off an open-source scalable vector graphics editor called InkScape v.1.3.2, which is used for moderate design work. The major advantage of our technique is that the design work can easily be conducted in traditional format (e.g., AutoCad v.22.0) and then uploaded to Inkscape to be printed. This allows for real-time revisions of microfluidic designs. Traditional masks used in photolithography can be costly and require a minimum lead time of 1–2 days for arrival. With our technique, designs can be created in AutoCad v. 22.0 and almost instantly printed on a substrate. This allows for rapid prototyping of a microfluidic device at a sub-one-dollar price point. It should be noted that Inkscape is vector-based graphics, meaning that curved lines are essentially small cubes. The AxiDraw printing technique discussed in the system optimization works to minimize the rigid feature structure, creating smooth curves. 

## 3. System Analysis

Initial optimization of the calligraphy system was performed to allow for increased performance when creating the microfluidic channels. Performance was based upon line smoothness, height, uniformity, and width. Optimization parameters included the following factors: temperature of the extruder tip, height of the tip from the substrate surface, write velocity, substrate temperature, and angle of the extruder tip with respect to the substrate. 

### 3.1. Extruder Tip Size

The system can exchange between a range of kistka tips ranging in inner diameters of the extruding orifice sizes: 3× Fine (254.0 μm), 2× Fine (330.0 μm), Extra Fine (457.2 μm), Fine (533.4 μm), Medium (660.6 μm), and Heavy (889.0 μm). This grants the ability to print features with different heights, thicknesses, and widths. It should be noted that each tip has the same reservoir volume, meaning that the only variable amongst tips is the extruder nozzle size located at the end of each tip. The tips can be seen in Figure 4, where they were imaged under optical magnification. Each tip is made of aluminum, allowing for high fluctuations in heat. The heat is controlled via a voltage regulator, allowing for temperatures ranging from 60 °C to 72 °C. As seen in Figure 5, varying tips result in a multitude of different values. These data were collected via a room-temperature slide (23 °C) at a maximum tip heat of 72 °C with a tip angle of 80° relative to a glass substrate. It should be noted that these values can be optimized for fluidic channel width and height based upon operating parameters, as later discussed.

Each tip was then used to print a single wax line on a glass substrate. The profile of each line was then measured using a surface profiler (KLA Tencor Alpha Step-IQ, CA, USA), as shown in Figure 5. The six tips’ profiles were compared in terms of channel width, height, and channel shape. In addition, the quantified data of the various tips ranges from the extruder tip inner diameter to the corresponding max channel width and height. As expected, as the extruder tip diameter decreased, the aspect ratios decreased. However, there was also a decrease in geometric uniformity as the tip size decreased. As will be discussed in Section 5, channel geometry has a large impact on fluidic flow and resistance in a microfluidic chip. This proved to be a baseline experiment to test the initial capabilities of the system. Keep in mind that the optimization and advancement of this system entail obtaining smaller feature sizes and increased channel uniformity.

### 3.2. Write Speed and Angle 

First, the optimization parameters tested were those of writing speed and angle or orientation with respect to the wax height on the substrate. The writing speed is increased from 2.0 mm/s to 265.2 mm/s (1% to 40%) of maximum machine capabilities. There was a direct correlation between higher writing speed and non-uniformity in the lines. A plateau effect occurs with any lines that are faster than 40.0 mm/s. These lines begin to level and have the same heights and widths. This allows us to control the parameters of the line anywhere between 2.0 mm/s and 80.0 mm/s. The angle of the tip has a large effect on the structure of the lines. Lines drawn between the speeds of 2.0–20.0 mm/s are much more uniform in the sense that they create a uniform profile of a half circle. The relationship between the velocity and the channel height can be seen in Figure 6, where the angle of the writing tip with respect to the substrate was altered. The initial angles tested were 22.5°, 17.5°, 12.5°, and 7.5° with respect to the normal. The process of decreasing the angle of the tip results in lines that are higher in height since the tip no longer compresses the wax as it flows out. However, as the write speed increases, there is a significant decrease in the channel height. This can be attributed to the fact that this technique uses a contact-based printing method, meaning that, at higher write speeds, the wax flow cannot be maintained due to such a large volume of wax being pulled out of the tip. 

### 3.3. Heating and Cooling of Substrate

To control line thickness, smoothness, and uniformity, the temperature of the glass substrate was altered before wax deposition. Substrate temperature has a significant impact on the wax printing technique for several reasons. As this is a contact-based printing method, the heat transfer coefficients of the materials affect the wax flow properties. If the substrate is too cold, the wax will not flow, and no channels will be created. Inversely, if the substrate temperature is too high, the wax will remain in its low-viscous form and flow outwards on the substrates. This results in the large spreading of the wax and, thus, low resolution and sharp lines. Optimization of the substrate temperature allows for smaller feature sizes as well as increased uniformity. As shown by Kaigala, heating the wax pattern post-deposition allows for a slight reflow of the outer surfaces of the wax channels, smoothing the profile [32]. Reversely, optimizing the temperature of the substrate cools the wax at a faster rate, allowing for less spreading and thus smaller feature sizes.

A visual representation of the wax line width and height as a function of temperature can be seen in Figure 7. Here, the substrate temperature was altered from 16.8 °C to 31.2 °C. Wax lines were drawn on individual glass substrates at a write velocity of 0.2 cm/s and a tip angle of 80° with respect to the substrate. Lines were drawn onto the glass slide using an extra-fine tip (taking note that there is a 3× finer tip that yields the smallest channel lines). These lines were then measured at three different points, and the average values and standard deviations were recorded (Figure 7b). The results showed that at the coolest temperature, the line height and width were the smallest, 80.25 μm and 346.21 μm, respectively. However, channel profile and uniformity decreased, as shown in Figure 7c. As shown by comparison to lines drawn at room temperature, applying heat and cold decreased and increased the uniformity in the lines geometric profiles, respectively. This suggests that to obtain smaller feature sizes and increased uniformity, a cool substrate should be used, followed by slight heating to remove non-uniformities in the channel profiles. From a physics standpoint, the temperature of the substrate affects the viscosity and thus the wax flow distance from the tip. This relationship can be seen in Equation (1), which shows the spreading of the wax as a function of viscosity and as a function of temperature. Further works include reflow heating optimization to further increase wax profile uniformity.

### 3.4. Software-Based Optimization 

Within the Inkscape software, customizable parameters are available to optimize performance and resolution. A curve smoothing function was implemented to adjust the precision with which the curves are interpolated for plotting. This essentially adjusts for the step size and write speed of the design while printing curved objects. The second technique deals with the cornering speed factor. This factor controls how much the machine slows down while going around a corner. A lower factor results in smooth curves, which is beneficial for spirals or circles. Finally, the steps in which the motors translate from positions were adjusted to a fine resolution, more specifically, 2000 DPI, giving smoother, neater plots. Figure 8 is a set of spirals created using the software optimization methods. These spirals were created using a 3× fine wax extruder tip, 1% write speed, a cornering factor of 1, and an angle of orientation of 78.5° with respect to the substrate. An extensive set of experiments was conducted on the optimization of this mechanical system, targeting the wax printing optimization of numerous geometric patterns that are commonly seen in microfluidic applications. A more detailed approach to this optimization is demonstrated in the works of N. Eadie [33]. The results of this optimization can be seen in Figure 8a. The unoptimized fluidic spiral mixer exhibits rough, jagged wax deposits, while the optimized version (Figure 8b) shows much higher uniformity and smoother features. The fluidic device was created using the PDMS fabrication technique, as mentioned in Figure 9.

### 3.5. Substrate Material 

This system allows the printing of wax channels on multiple substrate materials, including, but not limited to, copper, glass, plastic, acrylic, paper, and wood. This allows for the ability to translate the technique into multiple different applications requiring different materials. Traditional wax microfluidics are patterned mechanically on the paper by an ink jet printer. In this method, the wax spreads on the paper due to the porosity of the paper, as demonstrated by Washburn’s Equation (1) [19,34].
(1)$$L^{2}=\frac{\gamma Dt}{4\eta }\;$$
where *t* is the time for a liquid of dynamic viscosity *η* and surface tension *γ* to penetrate a distance *L* into the capillary whose pore diameter is *D*. As opposed to paper, glass has a lower capillary pore diameter and surface tension, meaning that there will be less spreading during the wax deposition process, yielding a higher channel resolution and smaller feature size.

Here, a series of materials were printed on the substrate, and the wax line width was measured. These tests were performed using a write speed of 2.0 mm/s, a tip angle of 80 degrees, and a 3× fine tip. As shown in Table 2, next to copper, glass yielded the smallest width. This holds true as the surface tension of glass is approximately 250–500 dyne/cm [35], while typical plastic falls between the 30–40 dyne/cm range [36]. Paper and wood, being porous materials, resulted in the most spreading of the wax, yielding larger widths. Sequentially, copper, which has a surface tension of 1000 dyne/cm, yielded the thinnest channel size. It should be noted, however, that the angle of the tip for writing on copper had to be adjusted to 72.5°, which is expected to affect the channel size, the extent of which is not known. 

This technique grants the ability to write on metals like nickel, gold, chromium, and/or copper, providing a possible application involving the integration of microfluidics into a flexible electrode [37]. It should be noted that the purpose of this optimization process was to understand how the parameters (writing speed, angle, cooling rate, etc.) affect the wax deposition and thus the characteristics of that channel (height, width, smoothness, uniformity). Meaning that ideally, based on the application of the device, a user would be able to adjust the system to reach the desired print results. 

**Table 2 micromachines-15-00240-t002:** A group of materials with measured channel width after printing and surface tension.

Material	Channel Width (µm)	Surface Tension (dyne/cm)
Copper *	180.6	1000 [35]
Glass	212.9	~250–500 [38]
Plastic	277.4	~30–50 [36,38]
Acrylic	316.1	35 [38]
Paper	322.6	-
Wood	380.6	~40–60 [39]

* This line required an angle of 72.5 degrees as compared to 80 degrees for the other lines. All measurements performed in stable room temperature of 26.6 °C and control humidity.

## 4. Device Fabrication Methods

The devices were fabricated using two separate techniques: a soft lithography approach [40] and the wax-based fabrication approach proposed here. The devices were then compared from a functional standpoint. Fabrication processes were compared to each other based on complexity in process steps, instrumentation start-up costs, per-device costs, fabrication time, and the quality of the device created.

### 4.1. Soft Lithography Fabrication 

The traditional soft/photo lithography fabrication technique is summarized in Figure 9. This technique is a common form of microfluidic fabrication. An overview of the process used to create the devices in this paper is as follows: Firstly, a silicon wafer is cleaned using a hydrofluoric acid bath for 10 min to remove the silicon dioxide layer on the wafer surface, after which the wafer is. The next photoresist (SU-8 2015, Micro-Chem Corp., Newton, MA, USA) was spin-coated on the wafer. A photomask, which needs to be designed and printed by a third party, is used in conjunction with an ultraviolet photolithography machine to etch the pattern onto the photoresist. The design is then developed. From here, a PDMS material was created using a pre-polymer and curing agent (Sylgard 184, Dow Corning Co., Midland, MI, USA) with a mixing ratio of 10:1. The mixture of the pre-polymer and the curing agent was poured on the etched wafer substrate and cured at 80 °C for 3 hrs. To create the actual device, the cured PDMS layer was peeled off, and then oxygen plasma was bonded to a glass slide using a PDC-32G (Harrick Plasma, Ithaca, NY, USA) at a power of 18 W. It should be noted that steps in Figure 9A(i–vii) were conducted in a class 1000 cleanroom environment.

### 4.2. Wax-Based Fabrication 

The wax-based fabrication method consisted of the following steps: First, a glass substrate was cleaned using an isopropanol rinse. From there, the slide is placed on top of the cooling bed for 5–7 min to allow the slide temperature to decrease. Then, wax is printed on the slide’s surface using the mechanical x-y actuation, based upon the CAD design file. From this point, the same fabrication steps of the soft lithography process are observed: PDMS is created and poured over the top, torn off, then plasma bonded to a clean glass slide (Figure 9B). 

In terms of fabricated device comparison criteria, i.e., channel uniformity and feature size resolution, traditional soft lithography shows higher capabilities. For instance, in terms of feature size, soft lithography can produce features 10 μm wide, while wax feature size is currently limited to features around 160 μm wide. Channel uniformity is also better in soft lithography; however, there are a number of system optimization techniques that can be applied to this wax-based technique, such as re-flow techniques that allow for increased uniformity. It should be noted that many of the microfluidic applications tested in this study showed close performance results when comparing the two fabrication techniques. 

A preliminary cost analysis was performed to compare the two techniques, first from an equipment start-up cost and second from a unit device cost. As shown in Appendix A, the proposed wax-based fabrication technique is significantly less expensive to not only set up but also operate. It eliminates the need for expensive hazardous chemicals such as hydrofluoric acid, photoresist, and developer. These are all chemicals that need to be replenished over time. From a unit device cost perspective, the costs are comparable. While traditional soft lithography has better resolution and uniformity than the wax-based technique, the initial equipment costs, continued costs of perishables, and operational complexity create a high barrier to entry for researchers to prototype designs and devices. At a sub-$1000.00 price point, the wax-based technique opens up microfluidic device fabrication to a larger number of researchers, students, and/or developing population groups.

## 5. Computational Model

As shown in previous studies (Schneider, 2018, p. 162) [41], in order to determine the ideal operating parameters for the desired application design specifications, a CFD model was created. Thus saving on financial and temporal costs. Through this model, the shape, flow, temperature change, and other physical properties are easily illustrated, which permits the ability to adjust and optimize. The software used for the modeling was the Flow-3D CFD V10 software package (www.flow3d.com, accessed on 3 July 2018). This model solves the Navier–Stokes equations using the volume of fluid (VOF) method [42] to track the free surface of the molten wax and the solidification of the wax based on porous media. The model implemented a solid fraction variable, which allowed for the solidification state to be determined within a fluid, in this case wax. The solid fraction can be described as the ratio of the solidified material with respect to the total amount of material in the computational domain. Once the solid fraction is below a critical value, a Darcy-type drag force is implemented in the Navier–Stokes equation.
(2)$$\rho \times v\left(\frac{\partial }{\partial t}+v\times \nabla \right)=-\nabla p+{\mu \nabla }^{2}v\;$$
where ρ and *µ* are the density and viscosity of the liquid wax, and *v* and *p* are the velocity and pressure. ∇ is the divergence. At points where the solid fraction is above the critical value, the drag is set to infinity, and the liquid behaves as a solid. To calculate the solid fraction, the model tracks the temperature of the wax and the relevant temperature fluxes via a thermo-fluidic heat transfer equation.
(3)$$\rho \times C_{p}\left(\frac{\partial T}{\partial t}+v\times \nabla T\right)={k\times \nabla }^{2}\times T\;$$
where *C_p_* is the specific heat of wax, *T* is the absolute temperature, and *k* is the thermal conductivity.

The computational domain (CD) for this model is shown in Figure 10. In this model, the lowest part of the extrusion tip is modeled by applying a hydrostatic pressure condition ⍴gh on the top boundary of the CD, which reduces computational expense. Here, g is gravitational acceleration, and h is the height of the fluid above the x-y plane of the CD. This pressure, driven by gravity, represents the remainder of the fluid above this small part of the nozzle and acts as the extrusion mechanism for the process. The CD is cut in half along the y-z plane, and a symmetry boundary condition is applied to reduce computational expense.

Temperature-dependent fluidic and thermal physical parameters were used for the analysis [43,44,45,46]. The major parameters used for paraffin wax were compiled from these sources into Table 1.

The beeswax was deposited directly onto a 20 °C glass substrate with a fixed position. The extrusion tip was moved at a constant velocity of 2 mm/s and tipped at an angle of 10 degrees from the normal to the substrate. A diagram of the simulation can be seen in Figure 11. The total simulation time was 1 s, resulting in a wax ridge 2 mm in length. To determine the accuracy of the wax channel shape and size of the computational model, the surface profile of the physically printed lines was measured using a profilometer and compared with the cross-sectional data from the flow model. The parameters used in these extrusions and simulations were the following: extruder angle from the surface normal = 10°, extruder tip velocity = 2 mm/s, wax temperature = 110 °C, and substrate temperature = 20 °C. The results, which are illustrated in Figure 12 and shown in Table 3, show an error of 1.18% in the difference in wax heights between the models and an error of 10.67% in the difference in cross-sectional area of the models. While most operating parameters were identical, some were not equivalent, such as external environmental conditions, where room temperature may have varied. Also, the temperature of the wax between the models differed, which caused a 3.5 mm^2^/s change in kinematic viscosity. From Washburn’s equation (Equation (1)), this temperature difference was likely the prominent reason for the error, as the solidification of wax directly changes with temperature.

## 6. Wax-Based Microfluidic Chip for Passive Micromixing

The concept of micromixing is an important function when dealing with microfluidic chips in order to realize the potential of these devices. Due to the inherent characteristics of microfluidics, such as a low Reynolds number, low laminar flow is typically seen in devices. This results in low fluidic turbulence and, therefore, low mixing of fluids. Research has shown this characteristic to be a hindrance to the use of microfluidic applications [14,47]. As a proof of concept for our wax-based microfluidic fabrication method, a series of three micromixers were created and tested: a spiral mixer, a rainbow mixer, and a serial dilutor. These applications were specifically chosen due to their ability to be both qualitatively and quantitatively compared. This was primarily conducted for two inherent reasons. The first is that these mixers can be viewed visually (both quantitatively and qualitatively). This allows for the comparison of the devices and their effectiveness in their applications. For instance, the process of mixing yellow and green dyes to make blue can be visually inspected. Diluting a solution at various concentrations, measured through luminescence, allowed us to validate the effectiveness of the wax microfabrication technique. The second reason is that because the original applications stem from validated traditional lithography approaches, we did not need to recreate that level of experimentation for comparison, allowing the focus to be on the wax fabrication technique and the comparison between traditional methods/performances as described in the cited research. 

The goal is to show that these wax-based microfluidic devices are comparable to those made using traditional photolithography fabrication methods. 

### 6.1. Spiral Micromixer

A spiral micromixer was fabricated using the proposed wax printing method; see Figure 8. The wax printing parameters consisted of a write speed of 80 steps/mm, a curve smoothing function of 100 (maximum software smoothness), a cornering factor of 1, a 3× fine kistka tip angled at 85° with respect to the glass substrate, and a wax temperature of 72 °C. The total print time of the device is 4 min and 53 s. Traditional PDMS fabrication methods were used, as mentioned. Two inlet holes of diameter 1 mm and an outlet of diameter 2 mm were punched out. An average channel width and height of 150 µm by 10 µm were achieved, respectively. Two dyes, one blue and one yellow, were injected simultaneously via dual syringe pumps at a rate of 30 µL/h using a dual syringe pump (KD Scientific Programmable Syringe Pump, Holliston, MA, USA). The initial pressure difference inside the syringes caused a pressure difference inside the device. This resulted in the blue dye backflowing into the yellow channel. Once equilibrium was reached, the dye flow was uniform. During the first few windings of the spiral, there was still a distinct interface between yellow and blue dye. Complete mixing began approximately halfway through the mixer, resulting in a green dye output (see Figure 8). 

### 6.2. Rainbow Mixer

Inspired by A. Grime’s “Shrinky-Dink microfluidics” [48], a microfluidic mixer was created in order to mix three different input dyes, red, blue, and yellow, at various concentrations, resulting in a fluid gradient containing the colors of the rainbow. The device was designed in AutoCAD and imported into Inkscape as an STL file. The time to print one device was 53 min. Traditional PDMS fabrication methods were used, as mentioned. Figure 13 is an image of the fabricated rainbow mixer. The wax drawing parameters included a write speed of 0.1326 cm/s, a curve smoothing function of 100 (maximum software smoothness), a cornering factor of 1, a 3× fine kistka tip angled at 10° with respect to the glass substrate, and a wax temperature of 72 °C. Three inlet holes of diameter 1 mm and an outlet of diameter 5 mm were punched out. Five flow channels connecting the outlet hole were incorporated into the device for the purpose of relieving fluidic pressure within the final mixing chamber, allowing for continuous flow of input fluids. Three dyes, blue/red/yellow, were injected simultaneously via syringe pumps at a rate of 5 µL/h using the dual syringe pump. Upon visual inspection, it is clear that a full spectrum of colors can be seen within the final mixing chamber. It should be noted that another capability being investigated is the ability to switch printing tips mid-draw to allow for different size channels as well as more uniform infill of large surface area structures. An example of where this would be useful can be seen on the bottom row of the rainbow mixer. The individual lines can be converted into one larger wax line, allowing for uniform fluid flow.

### 6.3. Serial Dilution Device

Serial dilution is the stepwise dilution of a substance in a solution. By controlling the lengths of the channels, it is possible to adjust and control the flow rates, which in turn are proportionally related to the fluidic resistance in the channel [49]. A general equation of flow resistance (RF) in channels can be modeled by the following equation:(4)$$R_{F}=C_{Geometry}\eta \frac{L}{A^{2}}\;$$
where *C_Geometry_* is the geometrical coefficient, *η* is the flow viscosity, *L* is the channel length, and *A* is the cross-sectional area of the channel. In the case of the linear dilutor discussed in et al. K. Lee [50], the input flow rate was determined by the aspect ratio of the fluidic channels. This comparison test between traditional photolithographic fluidic channels and wax-based fluidic channels aims to prove the geometric functions are comparable. In the original photolithographic linear dilutor, the channel sizes were 150 μm wide by 150 μm high, creating a specific fluidic resistance for that device. The length of the wax channels was created in proportion to the prior device, creating relative fluidic resistance amongst both devices and therefore the same dilution rates. The final device is shown in Figure 14. 

Channel geometry, and thus uniformity, is a major attribution for the fluidic resistance of a device. As such, the channel widths and heights at various points of the serial dilutor were measured using an Alpha-Step IQ surface profiler and the wax channel widths and heights were measured. The location of the measurements can be seen in Figure 15. Testing consisted of measuring two devices, one printed at a writing angle of 85° and one printed at 90°. Five adjacent profile measurements were taken at each site; the heights and widths were then averaged. The sites consisted of a “T” Junction, a Large “S” curve both the upper and lower halves, a small “S” curve both Upper and Lower half, and a straight line. It should be noted that the “S” curved contained an upper and lower half because, due to the difference in contact angle between the tip and glass when drawing right or left, the uniformity of wax line differs. The results shown in Figure 15 show that the average channel height and width printed at a writing tip angle of 85° with respect to the substrate were 34.22 μm and 227.80 μm, with a standard deviation of 12.27 μm and 45.35 μm, respectively. Comparatively, when the extruder tip angle is at 90° (i.e., perpendicular) with respect to the substrate, the heights and widths were 278.06 μm and 12.33 μm, with a standard deviation of 1.50 μm and 15.62 μm, respectively. What these data represent are channel uniformity. In this case, when the extruder tip is at an angle, the uniformity of the wax channels decreases. This is due to the fact that since this is a contact printing method, the extruder tip contact angle to the glass substrate is different depending on the direction of the movement. It should also be noted that the system at this point was unoptimized in the sense that there were limiting operating parameters that prevented uniformity in the Smaller Upper and Lower sections of the design. If these two data points are excluded, the standard deviation between lines drops by a factor of eight. This gives cause to believe that through more refined system optimization, higher uniformity amongst channels can be seen. However, with a tip angle of 90°, the results should have much better uniformity. Again, when the smallest features are excluded, the standard deviation decreases by almost half.

The experimentation for this device consisted of first running through two colored dye solutions and observing the color gradients after mixing. The second is the flow through fluorescent dye and a buffer solution, measuring the fluorescent intensities at the outlet. Using the identical wax printing method and parameters, a serial dilutor was printed on a glass substrate. The print time is 17 min. In both experiments, the fluids were pumped through a KD Scientific Programmable Syringe Pump for ten minutes before taking the dilution measurements. For the linear dilution using colored dyes, yellow and blue dyes were pumped into the inlets at a rate of 0.1 µL/h. After ten minutes, a gradient of color ranging from yellow to various shades of green to blue was seen near the outlet (see Figure 14). Theoretically, based on the fluidic circuit design of et al. K. Lee [50], a predictable linear dilution can be seen at each channel outlet. 

The second experiment consisted of flowing DI water and fluorescein sodium salt (1 mg/mL) solution into the buffer and sample inlets, respectively. After allotting for 10 min to allow for proper dilution, fluorescent images were acquired with a high-resolution monochrome digital camera (Hamamatsu ORCAER) mounted to an Olympus MVX10 epifluorescence microscope (Figure 16). Using the Olympus Wasabi imaging software package, a quantitative measurement of the fluorescent intensity at each dilution channel was obtained. 

## 7. Discussion

This method leaves the opportunity for a multitude of future works. Current experimentation is being performed with the modeling of the system through the simulation platform Flow-3D. Following the creation of the computational model of wax deposition with one set of parameters, further simulations can be achieved using expanded parameters. This updated model will be supplemented with experimentation using the physical testing of multiple design factors. Various substrates, such as silver, copper, silicon, and plastic, will be printed on to determine what substrate allows for the ideal channel size/shape. Substrate temperature will also be adjusted, as the spreading of wax varies directly with the surface temperature.

In an attempt to fully characterize the system and its operating parameters, a more intense optimization process is required. Through the implementation of a Taguchi Optimization technique, a series of test parameters will be evaluated based on the resulting channel uniformity and printing resolution. This technique tests parameters including substrate temperature, extruder tip temperature, writing speed, and writing angle. A Taguchi optimization technique allows for extensive coverage of parameters with minimal testing. The results of this technique can then be used to model our system in terms of equations dealing with the operating parameters. Manipulation of this equation will allow for a set of parameters that can correspond to a specific application need. Initial results have shown relationships between channel uniformity, channel design, and operating parameters. In terms of future experimentation, it should be noted that a combination of various parameters could be used during one print. Elaborated, with this type of wax printing system, a user can start printing with a high print speed, creating thin, small channels. Then, in real time, a user could adjust the print speed, angle, and/or platen temperature to make the channels taller or wider. This unlocks variable channel dimensions that could be beneficial for various microfluidic applications.

A third topic of discussion deals with the unique abilities of the fabrication process shown in Figure 9. This wax technique grants the ability to print, mold, and test a device all on the same slide without doing the tear-off process or plasma bonding, thus eliminating two steps and an industrial piece of equipment. To do this, the wax is printed on a glass slide as shown in steps (Figure 9B(i,ii)). However, now sacrificial wax pillars, a few millimeters high, are placed vertically on the inlet ports and outlet ports. PDMS is then poured on the glass slide and cured. However, now instead of tearing off the PDMS mold and plasma bonding it to a glass slide, the entire sample is placed in a wax-dissolving liquid such as acetone or mineral spirits. Upon use of a light vacuum, due to the inherent high porosity of PDMS, the PDMS will absorb the wax-dissolving liquid, and the inlet/outlet wax pillars, along with the design, will dissolve away, leaving behind a channel design.

This printing system is also compatible with paper substrates, as mentioned above in the discussion of paper-based microfluidics. Here, the wax printed is hydrophobic and can act as a barrier to a fluid channel. Initial tests have shown, like in previous studies, that this technique can yield smaller channel widths; however, for this study, the focus was on using the wax as a sacrificial layer.

Errors associated with this process play a critical role in the viability of the system and will be addressed to further improve system performance. This system was built as a proof of concept. As the initial prototype, there are a number of upgrades and modifications that will be made for version 2.0 that will increase the performance of the system, including line uniformity, print repeatability, ease of use, and line resolution. A few points of error lie within the non-uniformities of the substrate and the base of the acrylic board where the cooling/heating system was inset. This both affects the flow rate of the wax as well as the uniformity of the substrate temperature throughout the substrate medium. In addition, an investigation into various microtips are being conducted to further increase the feature size limit of the system. Finally, higher-resolution (steps/mm) motors is being investigated to further increase the vector printing limitations. As shown in Figure 16, the minor changes in geometric uniformity in channel widths and heights, effecting the fluid flow rate, cause the nonlinearity in the device output. Focusing on minimizing these errors will solve this problem. It should be noted that this system was designed to prove a new, untested technology and concept, and that these changes will only increase the performance of the fundamental methods shown in this paper.

## 8. Conclusions

This paper reports the development of a new, innovative, wax-based printing method to create microfluidic devices. This printing technology demonstrates a new pathway to rapid, cost-effective, device prototyping, eliminating the use of expensive micromachining equipment and chemicals. System analysis, considering design parameters such as extruder tip size, contact angle (0–22.5° with respect to the vertical), write speed (2.0–10 mm/s), substrate temperature (16.8–31.2 °C), and wax temperature, was used. Channels created ranged from 160 to 900 μm wide and 10 to 150 μm high based upon system operating parameters, set by the user to allow for system flexibility to accommodate a variety of applications. A computational model was created to show the approximate expectations of the wax channel shape, allowing for comparison with physical printed results. The wax height between the computational and experimental models was 57.982 μm and 57.306 μm, respectively, and the cross-sectional areas were 11,568 $\mu m^{2}$ and 12,951 $\mu m^{2}$, respectively. To validate system performance, a series of microfluidic mixers were created via the wax technique as well as photolithography: a spiral mixer, a rainbow mixer, and a linear serial dilutor. Results showed that while this technique was able to recreate traditional microfluidic devices within the known system resolution, there was a variance in the device’s performance. This means that, as the system stands in its current state, it is better used for prototyping and proof-of-concept purposes. Due to the inherent flexibility and cost advantages, as well as the closeness in device performance, this system has the potential to be used for low-cost, rapid microfluidic device prototyping.

## Figures and Tables

**Figure 1 micromachines-15-00240-f001:**
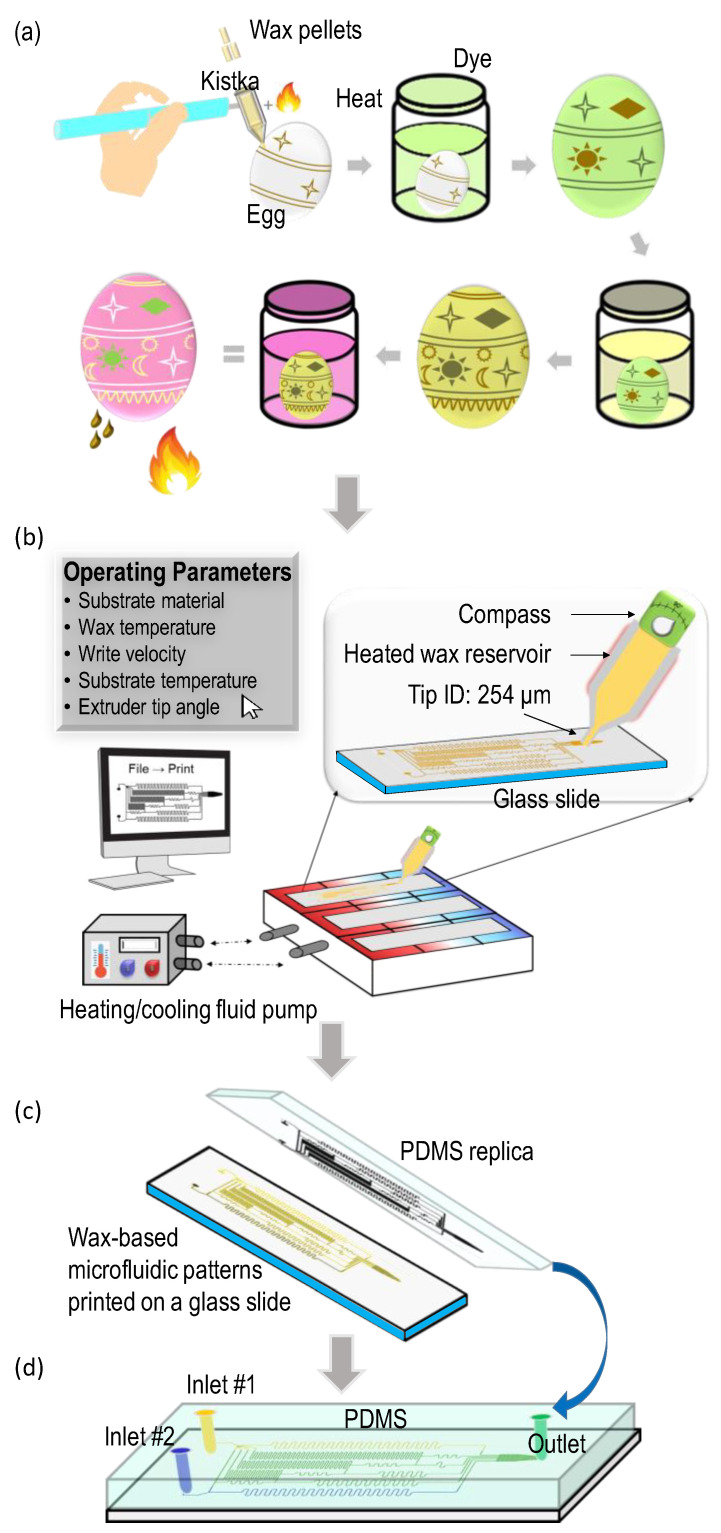
(**a**) The traditional Pysanky wax-based Easter egg painting technique. The tools and materials include an egg, candle, wax pallets, a Kistka, and dye. Wax pellets are inserted in the backend of the Kistka, a tiny metal funnel tip, and then the tip is heated over a candle. The melted wax begins to flow through the tip, and designs are drawn on the egg. The egg is dipped into dye, where the wax patterns resist the dye. This continues, and several different color dyes are used. Finally, the egg is warmed until the wax melts and is wiped off, revealing the colorfully patterned egg. (**b**) Automated wax-based contact printing with controllable operating parameters. The system contains a fluidic heating/cooling pump connected to the substrate platform, a wax extruder tip, a heated wax reservoir, and mechanical control of the print contact angle. (**c**) The microfluidic master mold printed on glass using the wax-based contact printing approach and the corresponding PDMS replica. (**d**) The PDMS microfluidic device is bonded to a new glass backing.

**Figure 2 micromachines-15-00240-f002:**
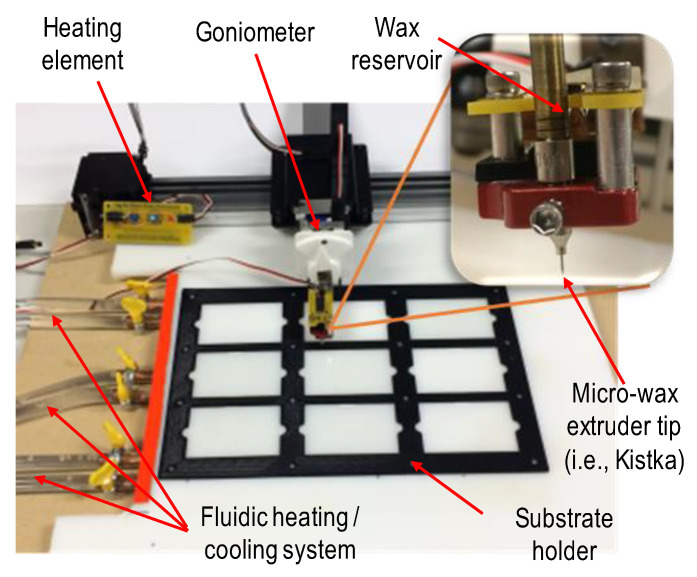
Wax printer technology consisting of X-Y mechanical actuation of a voltage-regulated heated aluminum extruder tip. A 3D-printed slide holder was mounted to an acrylic board. Underneath the board lies a fluidic network in which cold and hot fluids are pumped through. Cooling to prevent wax spreading, thus increasing the minimum channel feature size, and heating to allow for a re-flow of the wax to remove non-uniformities in the channel geometric profile.

**Figure 3 micromachines-15-00240-f003:**
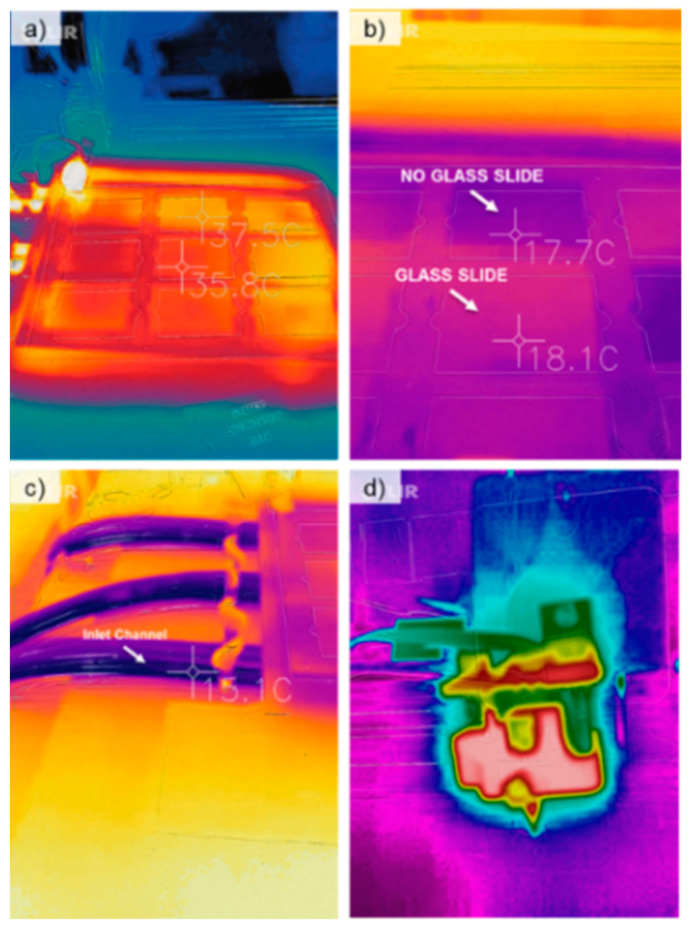
Thermal images taken of the system using the infrared camera. (**a**) The writing substrate after being heated for 20 min. (**b**) The writing substrate after being cooled for 20 min. The areas measured consisted of one with a glass slide and one without a glass slide. There is a 0.4 °C difference due to the heat transfer property of the slide. (**c**) The fluidic inlet channels of the system. (**d**) The wax extruder tip and heating elements show the heat concentration of the wax printing system.

**Figure 4 micromachines-15-00240-f004:**
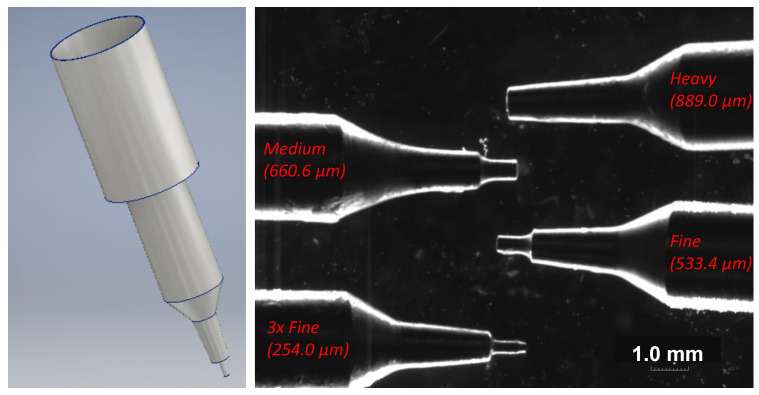
(**Left**) The CAD model wax kistka extruder tip used for the microfluidic wax printing. (**Right**) The optical microscope magnification of the various kistka tips available to extrude the wax and their respected inner diameter of the tip. Tip sizes ranged from 254 μm (3× Fine) to 889 μm (Heavy). Thin tip sizes were used to create the microfluidic patterns, while thicker tip sizes were used to fill, or shade, in areas.

**Figure 5 micromachines-15-00240-f005:**
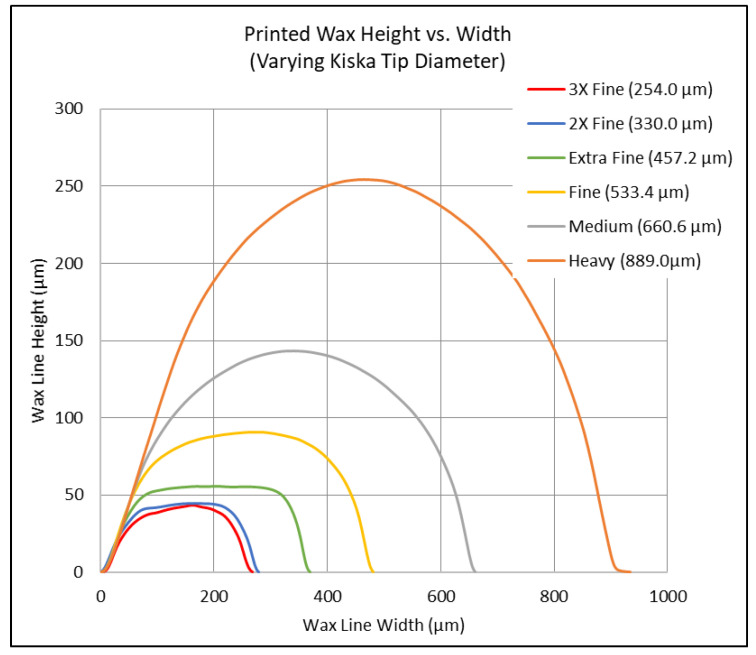
The line profile graph of each wax line printed using the various tip sizes. Widths ranged from 251.6 μm to 883.6 μm wide and 43.8 μm to 252.5 μm high. Operating parameters are a tip angle of 80° with respect to the substrate, a substrate at room temperature (23.6 °C), and a write speed of 2.0 mm/s.

**Figure 6 micromachines-15-00240-f006:**
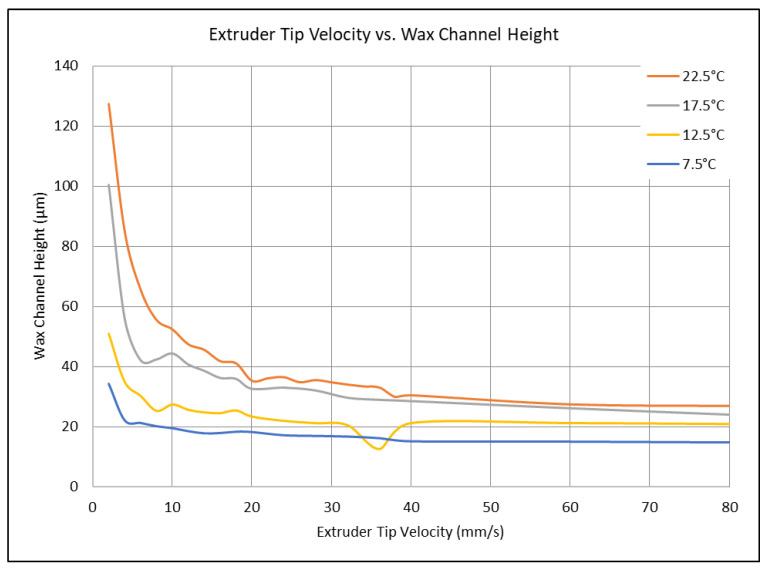
The extruder tip velocity versus the wax channel height. The correlation between write speed and extruder tip angle with respect to the substrate and wax line height can be seen. Printing conditions were performed at room temperature (23.6 °C), with a glass slide of 23.6 °C, using 100% pure beeswax and the extra-fine extruder tip size.

**Figure 7 micromachines-15-00240-f007:**
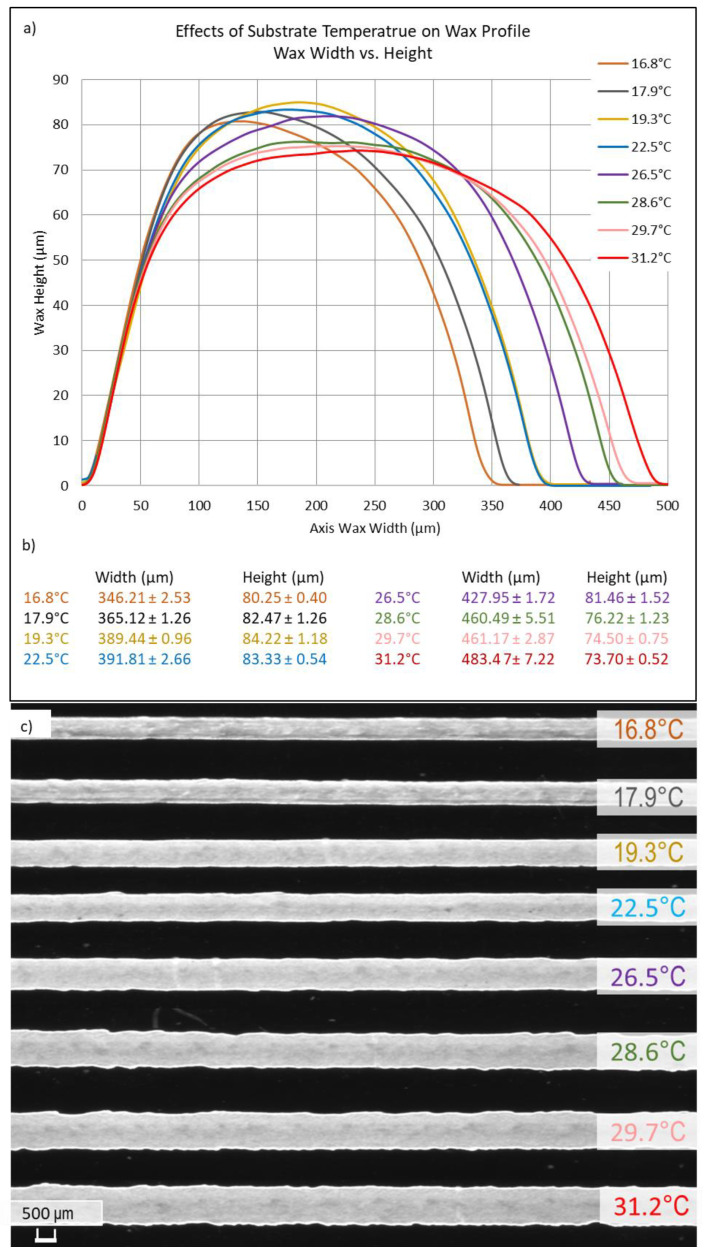
(**a**) Substitute temperature variation testing. Wax line geometric profiles as a function of the substrate temperature can be seen. The temperature ranged from 16.8 °C to 31.2 °C. (**b**) The averages and standard deviations of three measurement points on each line. (**c**) The microscope picture of each line drawn under its respected temperature.

**Figure 8 micromachines-15-00240-f008:**
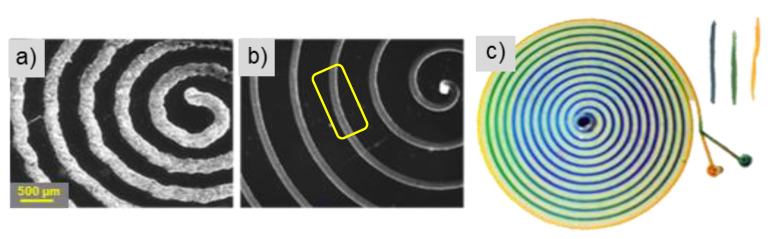
Fluidic spiral mixer utilizing software optimization methods for curved designs. (**a**) pre-optimization (**b**) post-optimization. Note that the yellow box signifies the area of the channel that was characterized using the profilometer. (**c**) Spiral mixing of yellow and green dyes results in blue. Print time of 4 min and 53 s.

**Figure 9 micromachines-15-00240-f009:**
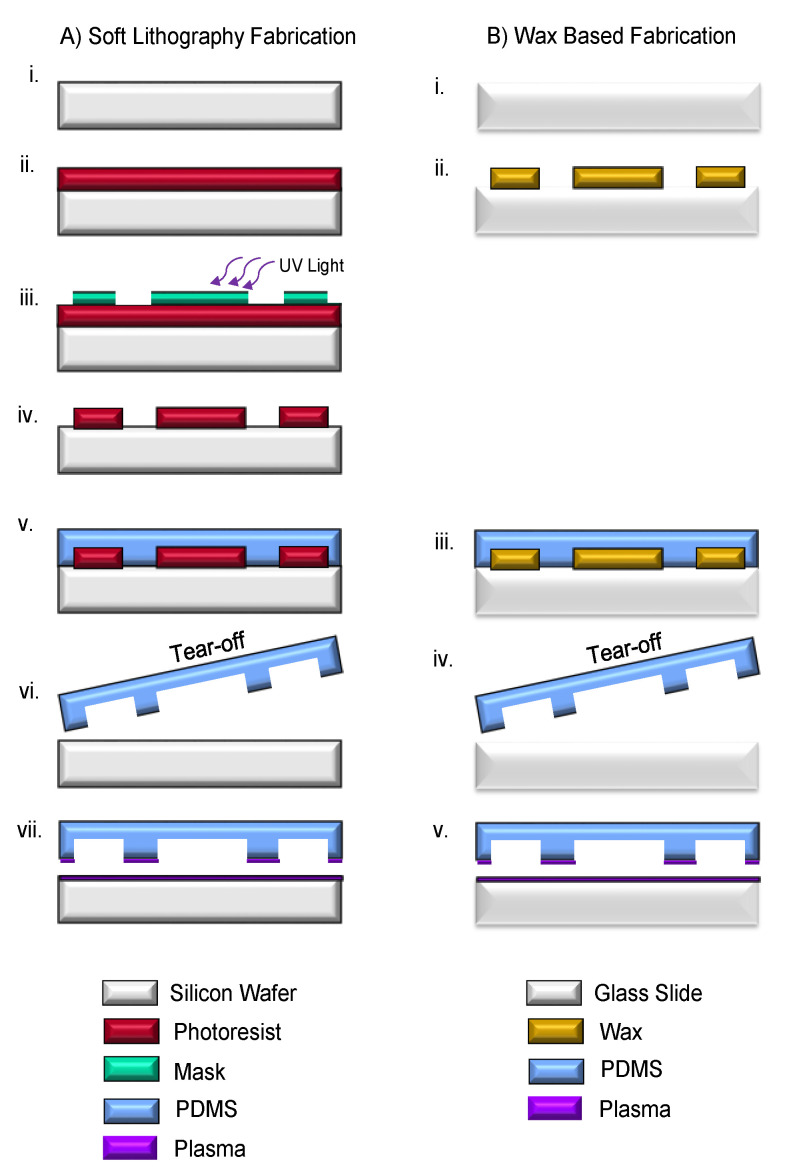
(**A**) The soft lithography fabrication process of the device. (i) The silicon wafer is prepped and cleaned using a hydrofluoric acid bath. (ii) Photoresist is spin-coated onto the wafer substrate. (iii) Photolithography and patterning photoresist through the designed photomask (iv) Development of SU-8 pattered on Si wafer. (v) PDMS (mixing ratio 10:1) on an etched wafer and cured at 100 °C for 15 min. (vi) PDMS is peeled from the substrate, resulting in a thin PDMS membrane, which serves as the microfluidic channels. (vii) Oxygen plasma bonding to a glass substrate. (**B**) wax-based fabrication method. (i) The glass substrate is prepped by rinsing with acetone, methanol, and isopropanol. (ii) Wax is deposited via the proposed wax printing system. (iii) The PDMS layer (10:1 mix ratio) is poured over the wax-glass substrate. (iv) PDMS is cured in a vacuum for 8 h then torn from glass. (v) PDMS with an imprinted design is plasma bonded to a clean glass slide, creating the final device.

**Figure 10 micromachines-15-00240-f010:**
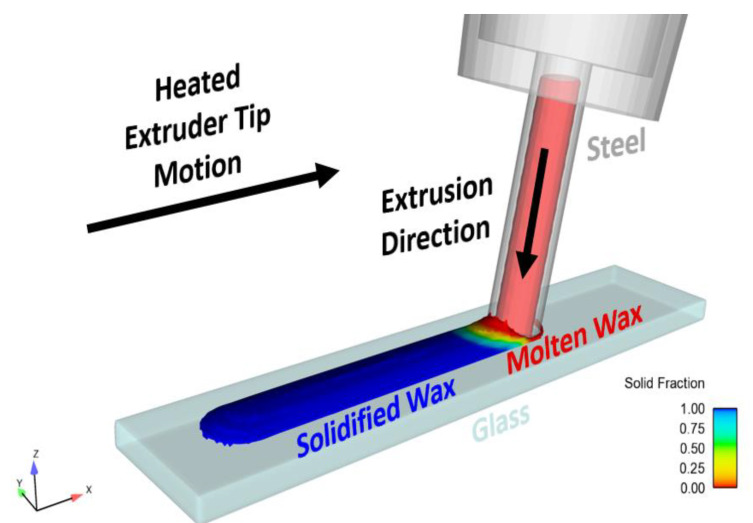
FLOW-3D thermofluidic simulation of the extrusion and solidification of beeswax during tip motion.

**Figure 11 micromachines-15-00240-f011:**
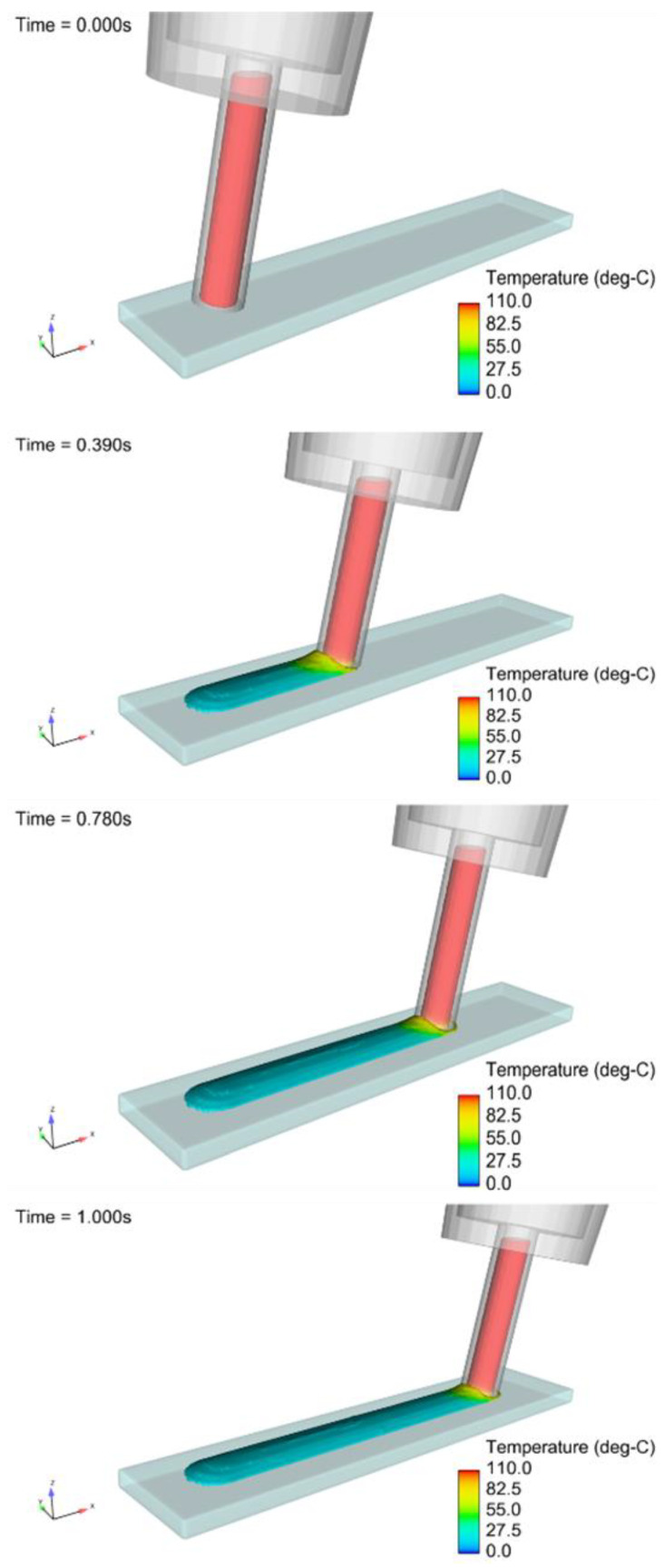
Time snapshots of the simulated temperature distributions of the beeswax ridge during printing.

**Figure 12 micromachines-15-00240-f012:**
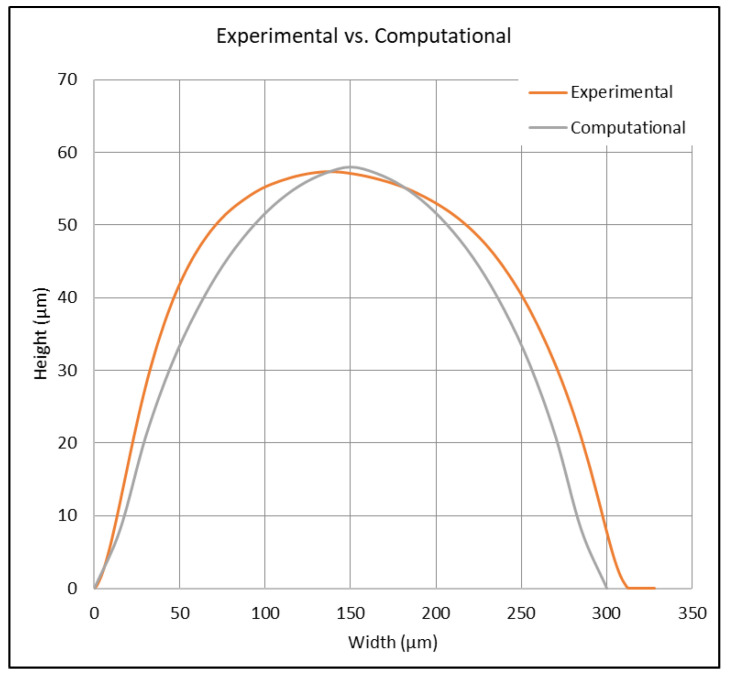
The comparison between computational and experimental results of the channel profile.

**Figure 13 micromachines-15-00240-f013:**
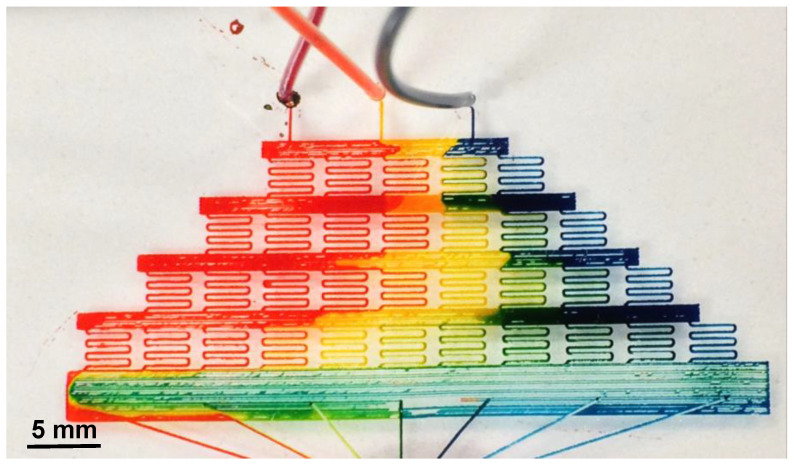
Rainbow mixer created via the wax printing method. The average channel width was measured to be 212 μm. Blue/red/yellow dyes were injected simultaneously at a rate of 5 µL/h. Upon visual inspection, a full spectrum of colors can be seen within the final mixing chambers. These colors represent the mixing effectiveness of the three primary colors of red, yellow, and blue as it propagates through the device.

**Figure 14 micromachines-15-00240-f014:**
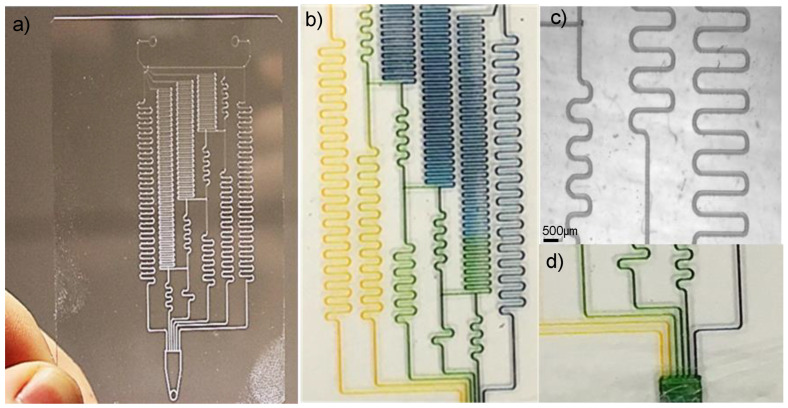
(**a**) The printed wax linear dilutor has a print time of 17 min, on a glass substrate. Note that the color dye is meant to be a visual representation of the mixing process. For example, yellow and blue mixed at various concentration results in various shades of green (**b**) Yellow and blue dyes are injected into the device, and the resulting color mixing. (**c**) The microscope image of the channels with an average channel width of 150 μm. (**d**) The viewing channels (**d**) of the device show yellow/blue dye ratios of 100/0, 80/20, 60/40, 40/60, 20/80, and 0/100.

**Figure 15 micromachines-15-00240-f015:**
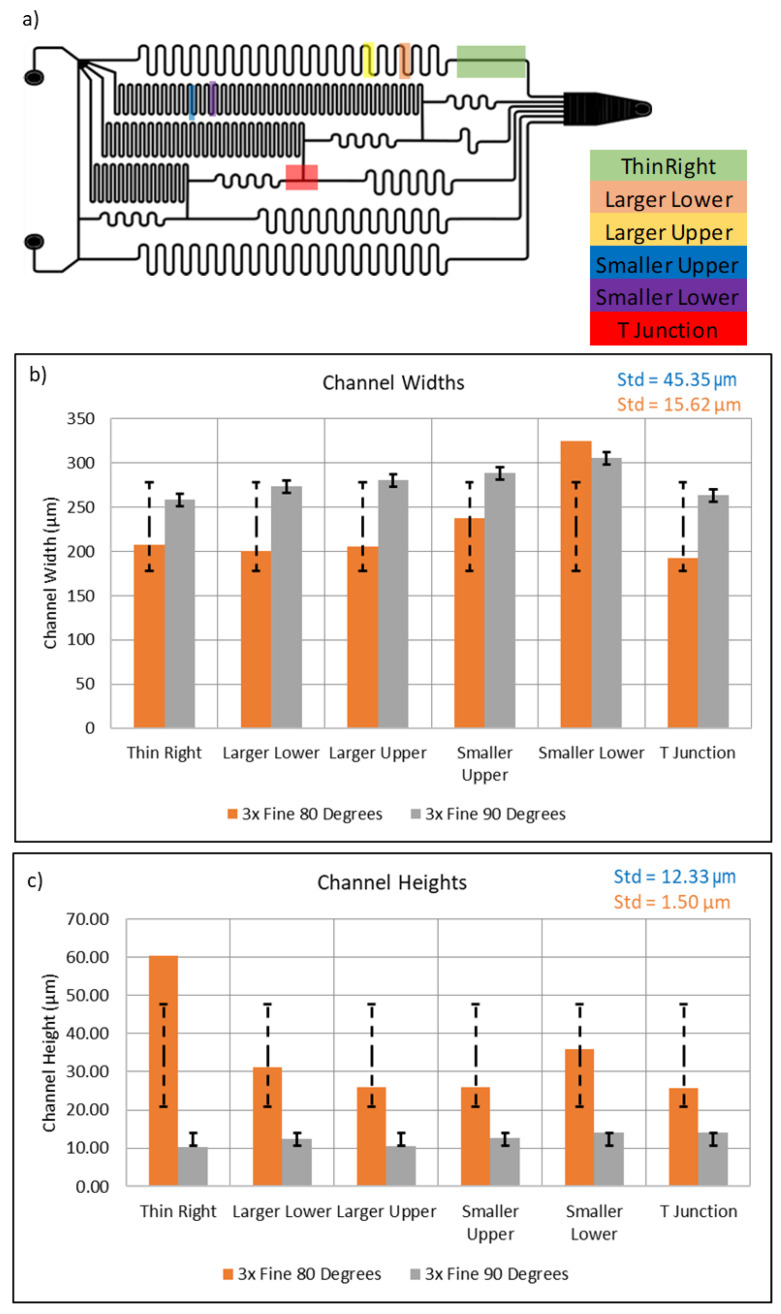
(**a**) The linear serial dilution device design with the area of examination heighted in colors. Areas of examination include Large “S” curve, both upper and lower halves; the Small Upper and Lower Halves, a “T” Junction” and a Thin Line. (**b**)The widths and (**c**) channel heights of each measurement location can be seen in the bar graphs above. Note that the deviation amongst measurements.

**Figure 16 micromachines-15-00240-f016:**
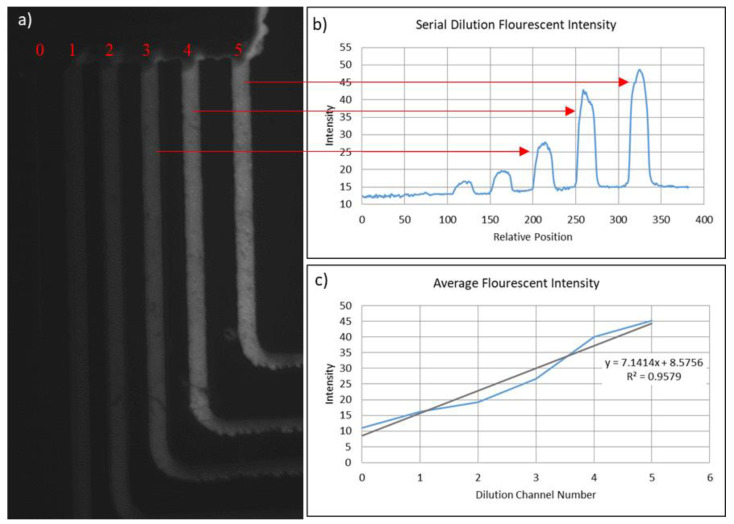
Fluorescent images acquired with a high-resolution monochrome digital camera (Hamamatsu ORCAER) mounted to an Olympus MVX10 epifluorescence microscope. (**a**) Theoretically, each dilution channel should increase in intensity by 20%, with 0% (far left) and 100% (far right). (**b**) The intensity of 10 points/channel was averaged and plotted, (**c**) the slope of which can be seen below. A best fit line was added for reference.

**Table 1 micromachines-15-00240-t001:** The physical properties of paraffin wax.

Property	Paraffin Wax
Melting Temperature	52 °C
Latent Heat	210 kJ/kg
Solid Density	860 kg/mm^3^
Liquid Density	780 kg/m^3^
Solid Specific Heat	2.9 kJ/kg K
Liquid Specific Heat	2.1 kJ/kg K
Solid Thermal Conductivity	0.24 W/m K
Liquid Thermal Conductivity	0.15 W/m K
Viscosity	0.205 Ns/m^2^

**Table 3 micromachines-15-00240-t003:** Computational model results compared to the physical model.

Parameter	Computational Model	Physical Model	Error (%)
Wax Height (µm)	57.98	57.31	1.18%
Cross-Sectional Area (µm^2^)	11,568	12,951	10.67

## Data Availability

Data are contained within the article and Appendix A.

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
