# Peer review of "Pysanky to Microfluidics: An Innovative Wax-Based Approach to Low Cost, Rapid Prototyping of Microfluidic Devices"

_micromachines, 2024, doi:10.3390/mi15020240_

Round 1
Reviewer 1 Report
Comments and Suggestions for Authors
Schneider et al. reported a wax-based mold fabrication method inspired by Pysanky. Authors optimized the performance of the printing parameters and achieved a resolution of ~160 um for the channel width. This is an interesting method to fabricate molds for microfluidic devices. I recommend the publication of this work after addressing the following concerns properly.
1. Please provide more details about the “wax” used in this work. What are its chemical and physical properties? I assume these are needed when performing the numerical simulation. For some reasons, the supplementary files are not accessible to the reviewer.
2. Authors compared the present method with the standard soft lithography method. However, it is more appropriate to compare this method with FDM 3D printing as both methods share similar working principles.
3. Line 451-451, should it be Fig.9 instead of Fig.8?
4. Please comment on the capability/potential of creating 3D structures (e.g., channels with different heights in the same device) using this method.
5. In the discussion section, it was mentioned that it is possible to create readily usable PDMS device without peeling off and bonding processes. Are there any experimental results to support his claim?
Reviewer 2 Report
Comments and Suggestions for Authors
This manuscript demonstrates a wax-based contact printing method combined with polydimethylsiloxane ( PDMS ) device fabrication techniques to create microfluidic devices. This work is very novel and demonstrates a new avenue for rapid, low-cost, device prototyping with potential applications in research and educational settings, as well as in developing regions that do not have the means and equipment to design, create, and test microfluidic chips. But a major revision was needed before acceptance.
1. The introduction states that the system has system advantages and that its feature size exceeds that of similar wax printing techniques, but no specific comparison of different wax printing techniques is made.
2. Parameter optimisation is all based on the uniformity and aspect ratio of the extruded straight wax line as a reference, but the channel often involves curved lines, such as spirals, are the optimised parameters consistent with straight lines?
3. When optimising the parameters, basically the glass substrate is considered as the base material, is it necessary to consider the different fluidity of wax on different materials?
4. In the optimisation of writing speed and angle, the results of different writing speeds and angles are described, but the optimised solution is not presented.
5. Part III mentions sticking a PDMS plasma onto a glass slide and then placing the whole sample in liquids that dissolve the wax, which then dissolves, leaving pre-determined channels. Does the fact that these liquids dissolve the wax also have an effect on PDMS?
6. Specific comparisons of performance results in microfluidic applications between this technique and microfluidic devices made by soft lithography fabrication techniques should be added.
7 If you have the simulation, the mesh quality, boundary conditions and set up should be presented.
Comments on the Quality of English LanguageNo comments
Reviewer 3 Report
Comments and Suggestions for Authors
The research article titled, "pysanky to microfluidics: An innovative wax-based approach to low cost, rapid prototyping of microfluidic devices', is a well-articulated research article. Although the concept of fabrication of moulds is a new concept, however, it is difficult to change worldwide labs to make a transition towards this kind of microfluidics technique. The conventional lithography techniques have been aced throughout the last two decades, which makes it difficult for this study to create a pathbreaking impact.
While the authors have validated the technology using 2-3 conventional designs, can the authors discuss further on the possibility of creating microvalves and micropumps using this technology.
This paper can be accepted for publication for this out-of-box approach and initiating a new approach of rapid prototyping microfluidic devices.
Round 2
Reviewer 1 Report
Comments and Suggestions for Authors
all my concerns have been addressed